# Outcome After 68Ga-PSMA-11 versus Choline PET-Based Salvage Radiotherapy in Patients with Biochemical Recurrence of Prostate Cancer: A Matched-Pair Analysis

**DOI:** 10.3390/cancers12113395

**Published:** 2020-11-16

**Authors:** Nina-Sophie Schmidt Hegemann, Paul Rogowski, Chukwuka Eze, Christian Schäfer, Christian Stief, Sebastian Lang, Simon Spohn, Rieke Steffens, Minglun Li, Christian Gratzke, Wolfgang Schultze-Seemann, Harun Ilhan, Wolfgang Peter Fendler, Peter Bartenstein, Ute Ganswindt, Alexander Buchner, Anca-Ligia Grosu, Claus Belka, Philipp Tobias Meyer, Simon Kirste, Constantinos Zamboglou

**Affiliations:** 1Department of Radiation Oncology, University Hospital, 81377 LMU Munich, Germany; Nina-Sophie.Hegemann@med.uni-muenchen.de (N.-S.S.H.); Paul.Rogowski@med.uni-muenchen.de (P.R.); Chukwuka.Eze@med.uni-muenchen.de (C.E.); christian.h.schaefer@gmail.com (C.S.); Rieke.Steffens@med.uni-muenchen.de (R.S.); Minglun.Li@med.uni-muenchen.de (M.L.); Claus.Belka@med.uni-muenchen.de (C.B.); 2Department of Urology, University Hospital, 81377 LMU Munich, Germany; Christian.Stief@med.uni-muenchen.de (C.S.); Alexander.Buchner@med.uni-muenchen.de (A.B.); 3Department of Radiation Oncology, Medical Center, Faculty of Medicine, University of Freiburg, 79106 Freiburg, Germany; sebastiang.lang@googlemail.com (S.L.); simon.spohn@uniklinik-freiburg.de (S.S.); anca.grosu@uniklinik-freiburg.de (A.-L.G.); simon.kirste@uniklinik-freiburg.de (S.K.); 4Department of Urology, Medical Center, Faculty of Medicine, University of Freiburg, 79106 Freiburg, Germany; christian.gratzke@uniklinik-freiburg.de (C.G.); sekretariat.schultze-seemann@uniklinik-freiburg.de (W.S.-S.); 5Department of Nuclear Medicine, University Hospital, 81377 LMU Munich, Germany; Harun.Ilhan@med.uni-muenchen.de (H.I.); Peter.Bartenstein@med.uni-muenchen.de (P.B.); 6Department of Nuclear Medicine, University of Duisburg-Essen, 47057 Essen, Germany; Wolfgang.Fendler@uk-essen.de; 7German Cancer Consortium (DKTK), University Hospital Essen, 45147 Essen, Germany; 8Department of Therapeutic Radiology and Oncology, Innsbruck Medical University, 6020 Innsbruck, Austria; ute.ganswindt@tirol-kliniken.at; 9German Cancer Consortium (DKTK), Partner Site Freiburg, 79106 Freiburg, Germany; 10German Cancer Consortium (DKTK), Partner Site Munich, 81377 Munich, Germany; 11Department of Nuclear Medicine, Medical Center, Faculty of Medicine, University of Freiburg, 79106 Freiburg, Germany; philipp.meyer@uniklinik-freiburg.de; 12Bertha-Ottenstein-Programme, Faculty of Medicine, University of Freiburg, 79106 Freiburg, Germany

**Keywords:** prostate cancer, salvage radiotherapy, PSMA PET/CT, choline PET/CT, matched-pairs analysis

## Abstract

**Simple Summary:**

In this bi-institutional analysis including 421 patients, we report an overall high biochemical-recurrence free survival rate (58% after three years) after positron-emission tomography (PET)-based salvage radiotherapy (sRT). Additionally, the strong prognostic effect of prostate specific antigen (PSA) level prior to sRT in this patient cohort was observed in multivariate regression analyses. Finally, patients who received staging with two different PET tracers for sRT guidance (cholin vs. prostate specific membrane antigen (PSMA) PET) were compared via propensity score matching and no significant differences in biochemical-recurrent free survival BRFS between the two tracers were observed.

**Abstract:**

The purpose of this analysis was primarily to analyze biochemical-recurrence free survival (BRFS) after positron emission tomography (PET)-guided salvage radiotherapy (sRT) in a large cohort, and to further compare BRFS after PSMA vs. choline PET/ computer tomography (CT)-based sRT. This retrospective analysis is based on 421 patients referred for PSMA or choline PET/CT after radical prostatectomy due to biochemically recurrent or persistent disease. BRFS (PSA: 0.2 ng/mL) was defined as the study endpoint. Cox regression analyses were performed to assess the impact of different clinical parameters on BRFS. Additionally, propensity score matching was performed to adjust patient cohorts (PSMA vs. choline PET/CT-based sRT). The median follow-up time was 30 months. BRFS at three years after sRT was 58%. In the multivariate analysis, only PSA before PET imaging and PSA before sRT were significantly associated with BRFS (*p* < 0.05). After propensity score matching, 272 patients were further analyzed; there was no significant difference in three-year BRFS between patients with PSMA PET-based vs. choline PET-based sRT (55% vs. 63%, *p* = 0.197). The present analysis confirmed the overall high BRFS rates after PET-based sRT and the strong prognostic effect of PSA level prior to sRT. PSMA PET-based sRT did not have superior BRFS rates when compared with choline PET-based sRT.

## 1. Introduction

Within the first five years after radical prostatectomy (RPE), almost one-third of prostate cancer (PCa) patients have a biochemical relapse (BR) [1]. In this oncologic setting, current guidelines recommend early salvage radiation therapy (sRT) of the former prostate bed with or without androgen deprivation therapy (ADT) [2].

In order to possibly increase long-term disease control after sRT, an individualization of the treatment strategy considering the results of hybrid positron emission tomography/computer tomography (PET/CT) scans is increasingly applied [3,4,5,6,7,8,9,10].

For many years, choline PET/CT imaging demonstrated the highest accuracy for staging and sRT guidance in patients with biochemical relapse after RPE. For example, D’Angellilo et al. [11] delivered up to 80 Gy to choline PET-positive lesions within the prostatic fossa and reported a three-year biochemical progression-free survival rate of 72.5% after sRT.

In the last few years, prostate-specific membrane antigen positron emission tomography/computed tomography (PSMA PET/CT) has rapidly evolved to be the new gold standard in staging patients with BR and is nowadays recommended in the current guidelines. Overall, impressive PCa detection rates between 57–78% at low prostate specific antigen (PSA) serum levels of 0.5–1 ng/mL have been recently reported in prospective studies [12,13,14,15]. In direct comparison to ^18^F-choline PET-CT, two retrospective analyses [16,17] showed a statistically significant higher PCa detection rate for PSMA PET/CT. This was further corroborated in a prospective comparison by Morigi et al., with detection rates of 50% vs. 12.5% for a PSA level of < 0.5 ng/mL and 69% vs. 31% for a PSA level of 0.5–2.0 ng/mL when performing PSMA and ^18^F-choline PET-CT in the same patients [18]. Consequently, the first studies on PSMA PET/CT reported an impact of PSMA PET-positive findings on the treatment decision process before sRT [19,20,21].

Nevertheless, it is unclear whether the individualization of sRT concepts based on the information derived from modern PET tracers equally leads to an improved biochemical relapse free survival (BRFS). Additionally, it is unknown whether the superior diagnostic performance of PSMA PET/CT over choline PET/CT translates into improved outcomes after sRT. Hitherto, PSMA PET is not approved for commercial use in the United States and many other countries, unlike choline which was approved by the U.S. Food and Drug Administration in 2012 under an investigational new drug application [22]. Thus, the purpose of this German bi-institutional retrospective analysis was primarily to analyze BRFS after PET-guided sRT in a large cohort of 421 patients and further to compare BRFS after PSMA vs. choline PET/CT-based sRT. To account for the retrospective nature of this study and to increase the comparability between the two treatment groups, a matched-pair analysis was conducted (patients with PSMA PET/CT vs. patients with choline PET/CT prior to sRT).

## 2. Materials and Methods

### 2.1. Patient Population

Two clinical databases (Freiburg and Munich) were screened for patients who received PET-guided sRT from January 2008 onwards until August 2018. Patients with metastatic disease at initial staging before surgery or with ongoing androgen deprivation therapy (ADT) at time of PET/CT were excluded. Overall, a total of 421 consecutive patients underwent PET/CT prior to sRT after RPE due to persistent or rising PSA at the departments of Radiation Oncology of the University Medical Centers of Freiburg and Munich. Two hundred and fifty-five patients had PSMA PET/CT scans and 166 patients had choline PET/CT scans. This retrospective analysis was performed in compliance with the principles of the Declaration of Helsinki and its subsequent amendments [23] and was approved by the local Ethics Committee of the respective Medical Faculties (approval number of University of Freiburg: 519/17; approval number of University of Munich: 17-765). The requirement to obtain informed consent was waived.

### 2.2. PET/CT Imaging

Imaging was performed in all patients either with ^18^F- or ^11^C-choline PET/CT (*n* = 166) or ^68^Ga-HBED-CC PSMA PET/CT (*n* = 255) before the initiation of sRT. Detailed descriptions of the imaging protocols used have been reported before [24,25]. PET/CT was interpreted by one nuclear medicine physician and one radiologist or by two nuclear medicine physicians in the sense of a clinical report-based analysis.

### 2.3. Treatment Application and Follow Up

Treatment management following PET scans was documented for each patient. sRT concepts were identical between the patients that received choline and PSMA PET regarding RT technique, treatment fields, administration of ADT and overall doses and have been previously published [24]. In brief: the prostatic fossa was irradiated with a median total dose of 66.6 Gy in 1.8–2 Gy (54.5–75.6 Gy). A dose escalation to the prostatic fossa was performed in 155/162 (96%) patients with PET-positive macroscopic lesions in the prostatic fossa. In case of PET-positive pelvic lymph nodes or positive lymph nodes in surgery specimen (pN1) status, pelvic lymphatic pathways were treated with a normo-fractionated overall dose of 45–50.4 Gy. At the discretion of the treating physician, the boost to the PET-positive lymph nodes was either delivered simultaneously or sequentially (*n* = 120; median dose 57.6 Gy in 1.8–2.2 Gy). PET imaging detected distant metastases in 44 patients (abdominal lymph nodes or bone metastases) and 31/44 (75%) patients received normo-fractionated RT with a median dose of 42.5 Gy (range: 30–64.7 Gy) to these metastases. ADT was recommended to every patient with evidence of PET-positive lesions for two years [2]. Patients with no PET-positive findings and PSA serum levels <0.7 ng/mL were normally treated with sRT alone. In case of elevated PSA levels, >0.7 ng/mL concomitant ADT application was recommended. Overall, 158/421 (38%) patients received ADT.

Follow-up time was defined as the interval in months between sRT and the last recorded PSA. Follow-up examination was first performed six weeks to three months after sRT and then every six to 12 months.

### 2.4. Statistical Analysis

BRFS (PSA ≤ 0.2 ng/mL) was defined as the study endpoint. For statistical analysis, SPSS Statistics 25 (IBM, New York, NY, USA) was used. Differences in continuous variables (not normally distributed) were evaluated with a Wilcoxon rank sum test. For categorical data, Fisher’s exact test and Chi-square test were used. Time-to-event data were calculated using the Kaplan–Meier method and compared using the log-rank test. Uni- and multivariate Cox regression analyses were used to identify predictors for BRFS after PET/CT-based sRT. Propensity score matching between the two PET tracer groups (PSMA vs. choline) was performed based on pre-PET PSA levels, pathological lymph node status, postoperative PSA persistence vs. PSA recurrence and National Comprehensive Cancer network (NCCN) risk groups and a matched patient cohort was created. A two-sided *p* value of < 0.05 was considered statistically significant. To account for the possible effect of ongoing ADT on PSA values during follow-up, additional analyses including only patients without ongoing ADT at least 3 months prior to last follow-up were conducted.

## 3. Results

### 3.1. Patient and Treatment Characteristics

Considering the entire cohort, patients had a median initial PSA (iPSA) of 10.3 ng/mL (range: 1.1–368) with mostly pT2 disease in suregery (36.8%) and predominantly Gleason score 7 (53.2%). Intermediate risk PCa according to NCCN [26] was most often observed (46.3%). Pathologic lymph nodes were evident in 23.5% of all patients at the time of radical prostatectomy and 40.6% of patients had persistent PSA after surgery. In comparison to patients with choline-based sRT, patients with PSMA PET/CT-based sRT had a significantly worse oncologic tumor status regarding iPSA, tumor stage, Gleason score, pathologic lymph nodes and percentage with postoperative PSA persistence. The median PSA before PET imaging was 0.69 (range: 0.07–40.1) for all patients with no significant differences between the two cohorts. Overall, PET-positive lesions were primarily detected within the prostatic fossa (38.5%), followed by PET-positive pelvic lymph nodes (31.6%) and distant metastases (10.5%). Thirty-nine percent of all patients were PET-negative. The detection rate of PET-positive lesions was significantly higher in the PSMA (65%) compared to the choline cohort (55%).

Median PSA before sRT was 0.69 (range: 0–40.1) for the entire cohort without any significant differences between the two cohorts. ADT was administered to 38% of all patients and pelvic lymphatic pathways were irradiated in 38%, with a boost to the pelvic lymph nodes in 29% of all patients. Patients in the PSMA cohort received ADT concomitantly with sRT significantly more often and sRT to pelvic lymphatic pathways significantly more often, as well as a boost to PET-positive pelvic lymph nodes. In total, the median dose to the prostatic fossa was higher in the choline cohort than in the PSMA cohort (70.2 Gy vs. 66.6 Gy, *p* < 0.001). Patient and treatment characteristics are provided in detail in Table 1 and Table 2.

### 3.2. BRFS for the Entire Cohort

After a median follow-up of 30 months (range: 1–107), 267 (63%) patients had no evidence of biochemical recurrence with a two- and three-year BRFS after sRT of 69% and 58%, respectively (Figure 1). Considering the entire cohort, 45 (10.7%) patients had ongoing ADT within the last three months prior to last follow-up (choline group: *n* = 7 and PSMA group: *n* = 38). Overall, BRFS after two (62% vs. 80%) and three years (53% vs. 66%) was significantly higher for patients with PET-negative findings compared to patients with PET-positive findings (*p* = 0.002) (Figure 2). Compared to patients with distant metastases, patients without distant metastases on PET imaging had a superior two- and three-year BRFS after sRT (72% vs. 38% and 63% vs. 29%). Patients without metastases on PET and favorable accompanying risk factors (no PSA persistence, PSA levels < 0.5 ng/mL prior to sRT/PET imaging and pN0 status) had two- and three-year BRFS rates of 75–80% and 67–68%, respectively.

Results of uni- and multivariate Cox regression analyses are presented in Table 3. Out of 17 clinical parameters, 12 remained statistically significant in the univariate analysis. In the multivariate analysis, only PSA before PET imaging and PSA before sRT remained significant as continuous and categorical variables.

Uni- and multivariate Cox regression analyses were also performed by excluding all patients with PSA persistence after surgery. Thus, only patients with biochemical-recurrent disease were considered (*n* = 250) and comparable results were obtained (PSA values before PET imaging or sRT are the only significant factors in multivariate analyses).

### 3.3. BRFS for Matched-Pair Analysis: Choline vs. PSMA PET-Based sRT

After propensity score matching, 272 patients (136 patients in each group) were further analyzed. Patient and treatment characteristics are presented in Appendix A and Appendix A. Patient groups were well-balanced regarding pre-PET PSA levels, NCCN risk groups, pN status and PSA persistence after surgery. Consistent with the entire cohort, patients in the PSMA PET group received ADT (38% vs. 20%) and sRT to the pelvic lymphatic pathways (38% vs. 24%) significantly more often, whereas the RT dose to the prostatic fossa was significantly higher in the choline PET group (median dose: 66.6 Gy vs. 70.2 Gy). After propensity score matching, there was no significant difference in two- and three-year BRFS between patients with PSMA PET-based vs. choline PET-based sRT (63% vs. 73% and 55% vs. 63%, *p* = 0.197, Figure 3). Fourteen of these 272 patients (5.1%) had ongoing ADT within the last three months prior to last follow-up (choline group: *n* = 5 and PSMA group: *n* = 9). Furthermore, no statistically significant differences between patients with PSMA PET-based vs. choline PET-based sRT were observed when considering only patients with persistent PSA levels (*p* = 0.343) or only patients with PSA relapse (*p* = 0.345) after surgery, respectively. Finally, only patients with biochemical-recurrent disease and without lymph nodes in surgery (pN0 status) were analyzed (*n* = 180), and again no statistically significant difference between patients with PSMA PET-based vs. choline PET-based sRT was observed (*p* = 0.448).

### 3.4. BRFS of Patients Without Ongoing ADT at Last Follow-Up

BRFS was further analyzed after the exclusion of patients with ongoing ADT within the last three months prior to last follow-up. Considering the entire cohort (*n* = 376), Cox regression analyses revealed comparable results (Appendix A). In the propensity matched cohort (*n* = 258), no significant difference in BRFS between patients with choline vs. PSMA based sRT was observed (*p* = 0.126).

## 4. Discussion

PET/CT imaging for PCa has revolutionized the postoperative radiation treatment of patients with PSA persistence or recurrence. For the first time, sRT is no longer based on the likelihoods of possible sites of relapse, but on the visualization of the area of recurrence. However, the outcome of PET-based sRT [27] has not yet been extensively studied. Thus, the primary aim of this study was to retrospectively assess the biochemical response rate after PET-based individualization of sRT in a large bi-institutional analysis. In total, 61% of patients had PET-positive findings triggering RT dose escalation to PET-positive regions, adaptation of RT volumes, and administration of ADT.

The two- and three-year BRFS (69% and 58%) in the present study are similar to numerous retrospective [28,29,30,31] and prospective [2,32] studies reporting on non-PET-guided sRT. Overall, notably, the head-to-head comparison of treatment outcomes between trials is hampered by variant proportions of patients with accompanying ADT (0–50%), with pelvic lymphatic pathway irradiation, with a wide range of pre-sRT PSA values (median: 0.4–0.72 ng/mL) and by different applied dose regimens (median dose: 64.8–66.6 Gy). Trials explicitly investigating the oncologic benefit of PSMA PET-based sRT are currently underway, but results are still pending (ClinicalTrials.gov identifier: NCT03582774) [33]. Until then, it remains unclear whether implementing PET/CT as an expensive and radiation-burden carrying modality improves the management of these patients.

Several prognostic factors, in particular initial PSA value, Gleason score, pT stage, PSA persistence, pathologically-positive lymph nodes, and pre-sRT PSA value have been introduced by previous studies for the prediction of BRFS after sRT [34,35]. In the univariate analysis, all these factors were significantly associated with BRFS in the present study. However, in the multivariate analysis, only PSA value prior to PET imaging and prior to sRT remained a significant prognostication of BRFS. These findings underline the importance to initiate sRT at low PSA levels (<0.5 ng/mL) as it is recommended in current guidelines [13,36]. A prospective multi-center study by Emmett et al. reported on three-year freedom from progression after PSMA PET-triaged management of patients with PSA recurrence after surgery. The authors observed that men with either a negative scan or PSMA-positive disease confined to the fossa had significantly higher three-year failure free survival rates compared to those with either pelvic nodal or distant metastatic disease [37]. Giovacchini et al. equally showed that positive findings in choline PET/CT prior to sRT are an independent predictor for worse PCa-specific survival [38]. These findings are confirmed in the present study: patients with PET-positive findings had significantly worse BRFS compared to patients without PET-positive findings (three-year BRFS 53% vs. 66%), even after the exclusion of patients with distant metastases on PET/CT for subgroup analyses. Additionally, only PET-positive pelvic lymph nodes and distant metastases had a significant impact on BRFS in the univariate analysis, whereas PET-positive local recurrence within the prostatic fossa had no such impact. Thus, a further intensification of therapy may be warranted for patients with PET-positive lymph nodes or distant metastases. This might possibly be resolved by either the administration of higher radiation doses to PET-positive regions [39], an enlargement of radiation volumes [40], or by adding more aggressive systemic therapy regimens like concomitant docetaxel or abiraterone to sRT [41,42]. However, it is worth mentioning that patients with no metastases on PET and favorable accompanying risk factors had high two- and three-year BRFS rates of >74% and >66%, respectively. A risk-adapted treatment approach regarding the initiation of ADT is in line with a secondary analysis of the multicenter Radiation Therapy Oncology Group (RTOG)9601 trial randomizing men to sRT with/without ADT [43]. In this analysis, pre-sRT PSA level was a prognostic biomarker for outcome after sRT with an overall survival benefit for the combination with ADT limited to patients with late sRT initiation with PSA level >0.6 ng/mL. Consequently, in the coming years, a further sRT individualization based on PET imaging and inherent patient risk factors is expected. In this context, future studies should also address whether radiomic features derived from PET images [44] may improve risk stratification models for patients undergoing PET-guided sRT.

The second aim of the present study was to investigate BRFS in patients after choline- vs. PSMA PET-guided sRT. For better comparability, a matched-pairs analysis was conducted with a high congruence of pre-treatment parameters between the two groups. Overall, a PET detection rate of 58% PET-positive findings was alike in both groups. However, PSMA PET/CT detected significantly more positive pelvic lymph nodes, leading to a significantly higher percentage of patients with concomitant ADT (20% vs. 38%) and sRT to pelvic lymphatic pathways (24% vs. 38%) in this cohort. On the other hand, the sRT dose to the prostatic fossa was significantly higher in the choline group, which might be explained by the higher detection rate of local recurrences within the prostatic fossa by choline PET (47% vs. 37%). A possible explanation for this observation could be the high bladder signal of the used ^68^Ga-HBED-CC PSMA tracer, which might hamper the correct detection of local recurrent disease. Despite the higher detection rate of PET-positive lesions at lower PSA levels, this did not translate into a higher BRFS in patients with PSMA PET-based vs. choline PET-based sRT after propensity score matching. A possible explanation for this finding is that the current intensification of treatment strategies based on PET findings (e.g., administration of ADT or additional treatment of pelvic lymphatic pathways) are so far not sufficient. Additionally, the retrospective design of our study could have led to a heterogeneous distribution of prognostically relevant clinical parameters between both patient groups, which may not have been accounted for by propensity score matching. However, we also compared choline- versus PSMA-based sRT only in patients with true biochemical recurrence after surgery (by excluding patients with PSA persistence and patients with pN+ status) and did not find any statistically significant differences.

Our study has several limitations mainly due to its retrospective character: the treatment protocols and the follow-up procedure were not identical for all patients. A further shortcoming of the present analysis that limits drawing premature conclusions is the relatively short follow-up due to the only recent availability of PSMA PET/CT scans. Hence, at best, our primary endpoint (BRFS) may serve as a surrogate marker for more relevant endpoints like PCa-specific survival or overall survival. In addition, due to the median follow-up of 30 months, a prolonged effect of testosterone suppression on PSA levels at last follow-up cannot be totally excluded. To account for this, an additional analysis including only patients without ongoing ADT at least three months prior to last follow-up was conducted and the previous findings were reproduced.

## 5. Conclusions

The present analysis retrospectively analyzed PET-based sRT in a large patient cohort in detail, confirming high overall BRFS rates and the strong prognostic effect of PSA level prior to sRT. Further, it was corroborated that PET results are highly predictive of BRFS in men undergoing sRT for biochemical persistence or recurrence after radical prostatectomy, with overall lower BRFS in patients with PET-positive disease not confined to the prostatic fossa. Interestingly, BRFS rates were independent of the PET tracer used. Thus, not surprisingly, PSMA and choline PET/CT are nowadays universally used for stratification and treatment individualization with the results of a currently recruiting phase III trial [33] on the benefit of PSMA PET/CT in sRT patients eagerly awaited. Future studies should assess whether further treatment intensification in patients with PET-positive pelvic lymph nodes with the superior detection rate of PSMA PET/CT will translate into improved clinical outcome after sRT.

## Figures and Tables

**Figure 1 cancers-12-03395-f001:**
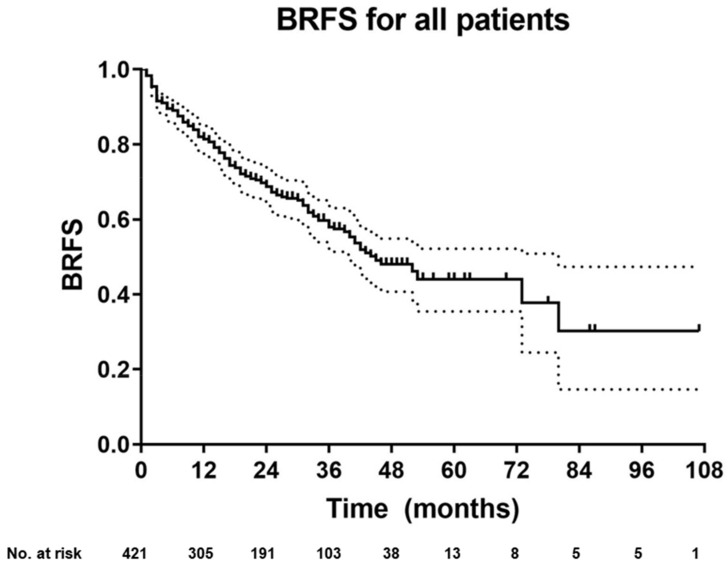
Kaplan–Meier curve of biochemical-recurrence free survival (BRFS) for the entire cohort. Plotted symbols represent censored points. Error bars show 95% confidence interval.

**Figure 2 cancers-12-03395-f002:**
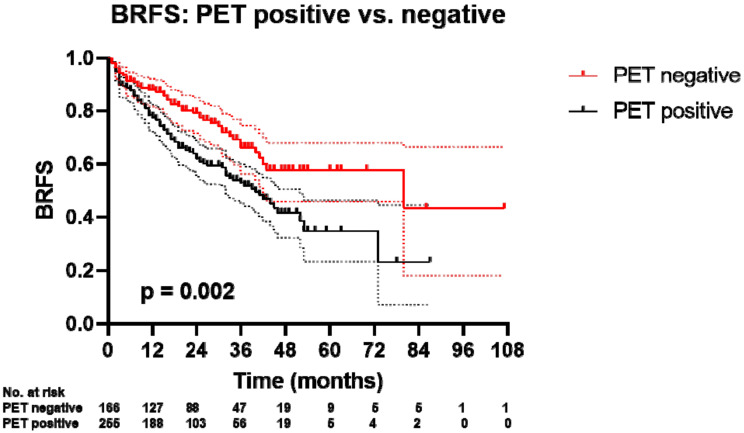
Kaplan–Meier curve of BRFS PET-positive vs. PET-negative patients. Plotted symbols represent censored points. Error bars show 95% confidence interval. *p* value was obtained with log-rank test.

**Figure 3 cancers-12-03395-f003:**
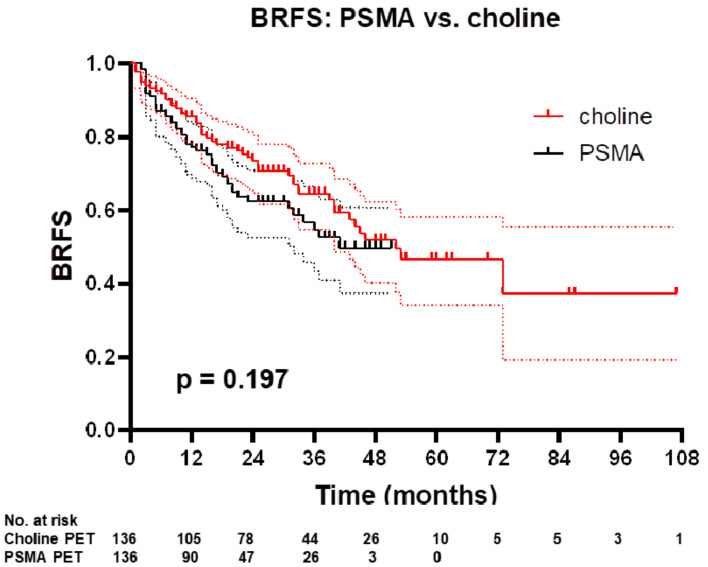
Kaplan–Meier curve of BRFS PSMA vs. choline PET in adjusted cohorts. Plotted symbols represent censored points. Error bars show 95% confidence interval. *p* value was obtained with log-rank test.

**Table 1 cancers-12-03395-t001:** Patient characteristics.

	Unadjusted			
	Total Cohort	Choline Cohort	PSMA Cohort	*p* Value
Number (%)	421 (100)	166 (39.4)	255 (60.6)	
Age in years (median, range)	69 (47–83)	69 (49–82)	71 (47–83)	n.s.
initial PSA in ng/mL (median, range)	10.3 (1.1–368)	9.6 (1.1–281)	11.1 (2.3–368)	**
pT stage (%)				****
2	155 (36.8)	59 (35.4)	96 (37.6)	
3a	107 (25.4)	43 (25.9)	64 (25.1)	
3b	117 (27.8)	31 (18.7)	86 (33.7)	
4	14 (3.3)	7 (4.2)	7 (2.7)	
unknown	28 (6.7)	26 (15.7)	2 (0.8)	
Gleason score (%)				****
6	27 (6.4)	16 (9.6)	11 (4.3)	
7a	107 (25.4)	48 (28.9)	59 (23.1)	
7b	117 (27.8)	50 (30.1)	67 (26.3)	
8	67 (15.9)	24 (14.5)	43 (16.9)	
≥9	87 (20.7)	17 (10.2)	70 (27.5)	
unknown	16 (3.8)	11 (6.6)	5 (2)	
NCCN risk group (%)				****
low risk	14 (3.3)	3 (1.8)	11 (4.3)	
intermediate risk	195 (46.3)	88 (53.0)	107 (42)	
high risk	193 (45.8)	59 (35.5)	134 (52.5)	
unknown	19 (4.5)	16 (9.6)	3 (1.2)	
R status (%)				****
R0	214 (50.8)	80 (48.2)	134 (52.5)	
R1 or R2	161 (38.2)	51 (30.7)	110 (43.1)	
Rx	46 (10.9)	35 (21.1)	11 (4.3)	
pN status (%)				***
pN0	283 (67.2)	127 (76.5)	156 (61.2)	
pN1	99 (23.5)	21 (12.7)	78 (30.6)	
pNx	34 (8.1)	13 (7.8)	21 (8.3)	
PSA persistence after surgery (%)				****
yes	171 (40.6)	46 (27.7)	125 (49)	
no	250 (59.4)	120 (72.3)	130 (51)	
PSA before imaging in ng/mL (range, median)	0.69 (0.07–40.1)	0.69 (0.07–39.0)	0.7 (0.13–40.1)	n.s.
PET findings (%)				****
Positive PET findings	257 (61)	92 (55.4)	165 (64.7)	
No PET findings	164 (39)	74 (44.6)	90 (35.3)	
Locally recurrent disease	162 (38.5)	77 (46.3)	85 (33.3)	
Pelvic lymph nodes	133 (31.6)	30 (18.1)	103 (40.4)	
Metastatic disease	44 (10.5)	10 (6.0)	34 (13.3)	
PSA before sRT in ng/mL (range, median)	0.69 (0–40.1)	0.7 (0–29.0)	0.66 (0–40.1)	n.s.

Abbreviations: n.s.: not significant, **: *p* < 0.01, ***: *p* < 0.001, ****: *p* < 0.0001, R: resection status, Rx: resection status unknown, pNx: pathological lymph node status unknown, PSA: prostate specific antigen, NCCN: National Comprehensive cancer Network, pN status: lymph nodes in surgery specimen, PET: positron-emission tomography, sRT: salvage radiotherapy.

**Table 2 cancers-12-03395-t002:** Treatment characteristics.

	Unadjusted			
	Total Cohort	Choline Cohort	PSMA Cohort	*p* Value
Number (%)	421 (100)	166 (39.4)	255 (60.6)	
ADT				****
yes	158 (37.5)	33 (19.9)	125 (49)	
no	263 (62.5)	133 (80.1)	130 (51)	
Duration ADT in months				****
≤6 months	87 (20.7)	26 (15.7)	61 (23.9)	
>6 months	54 (14.9)	2 (1.2)	52 (20.4)	
Unknown duration (*n*, %)	17 (3.9)	5 (3.0)	12 (4.7)	
RT dose prostatic fossa in Gy (median, range)	66.6 (54.5–75.6)	70.2 (56–72)	66.6 (54.5–75.6)	****
RT to elective pelvic lymphatics				****
yes	160 (38)	33 (19.9)	127 (49.8)	
no	261 (62)	133 (80.1)	128 (51.2)	
RT dose to elective pelvic lymphatics in Gy (median, range)	50.4 (36–54)	50.4 (36–54)	50.4 (45–52.8)	n.s.
RT boost to pelvic lymph nodes				****
yes	120 (28.5)	19 (11.4)	101 (39.6)	
no	301 (71.5)	147 (88.6)	154 (60.4)	
RT dose to pelvic lymph nodes in Gy (median, range)	57.6 (45–70)	56 (50.4–66)	58.2 (45–70)	n.s.
RT to distant metastases				n.s.
yes	31 (8.3)	10 (6.0)	21 (8.2)	
no	390 (92.6)	156 (94.0)	234 (91.8)	
RT dose to distant metastases in Gy (median, range)	42.5 (30–64.7)	35 (30–56)	45 (34–64.7)	n.s.
Follow-up time in months (median, range)	30 (1–107)	36 (1–108)	27 (1–56)	****
Biochemical recurrent disease				n.s.
yes	154 (36.6)	56 (33.7)	98 (38.4)	
no	267 (63.4)	119 (66.3)	157 (61.6)	

Abbreviations: n.s.: not significant, ****: *p* < 0.0001, PSMA: prostate specific membrane antigen, ADT: androgen deprivation therapy, RT: radiotherapy.

**Table 3 cancers-12-03395-t003:** Cox regression analysis for the entire cohort.

Univariate Cox Regression Analysis				
	Cox ratio (95% CI)	*p* value		
Indication RT (PSA persistence vs. salvage)	**1.7** **(1.2–2.3)**	**0.001**		
Risk group NCCN (low- vs. intermediate vs. high)	**1.4** **(1.1–1.9)**	**0.019**		
R status (R0 vs. R1)	0.75 (0.5–1)	0.092		
Gleason score (6 vs. 7 vs. 8)	**1.4** **(1–1.8)**	**0.023**		
pT (2 vs. 3 +4)	**1.6** **(1.2–2.3)**	**0.006**		
pN (0 vs. 1)	**1.6** **(1.2–2,1)**	**0.002**		
Time between surgery and PET imaging (days, continuous variable)	1.0 (1.000-1.000)	0.252		
Tracer (PSMA vs. cholin)	**1.5** **(1.1–2.1)**	**0.014**		
PSA before PET (</>0.5 ng/mL))	**1.7** **(1.2–2.4)**	**0.002**		
PSA before PET (ng/mL, continuous variable)	**1.06** **(1.04–1.09)**	**<0.001**		
PSA before sRT (</>0.5 ng/mL)	**1.7** **(1.2–2.4)**	**0.002**		
PSA before sRT (ng/mL, continuous variable)	**1.07** **(1.04–1.09)**	**<0.001**		
Initial PSA (ng/mL, continuous variable)	**1.005** **(1.001–1.008)**	**0.004**		
PET-positive findings (yes vs. no)	**1.7** **(1.2–2.4)**	**0.002**		
Local recurrence in PET (yes vs. no)	1.1 (0.8–1.5)	0.661		
Pelvic lymph nodes in PET (yes vs. no)	**1.8** **(1.3–2.5)**	**<0.001**		
Distant metastases in PET (yes vs. no)	**2.4** **(1.6–3.7)**	**<0.001**		
RT dose prostatic fossa (</> 66 Gy)	1.3 (0.7–2.2)	0.365		
ADT (yes vs. no)	1.08 (0.8–1.5)	0.632		
Multivariate Cox regression analysis				
	Cox ratio	95% CI		*p* value
		lower	upper	
Indication RT (PSA persistence vs. salvage)	1.154	0.774	1.719	0.482
Risk group NCCN (LR vs. IR vs. HR)	0.893	0.553	1.441	0.642
Gleason score (6 vs. 7 vs. 8)	1.072	0.686	1.676	0.760
pT (2 vs. 3 +4)	1.258	0.811	1.950	0.305
pN (0 vs. 1)	1.368	0.964	1.943	0.080
Tracer (PSMA vs. cholin)	1.019	0.685	1.517	0.926
PSA before PET (ng/mL, continuous variable)	**1.036**	**1.005**	**1.069**	**0.024**
Initial PSA (ng/mL, continuous variable)	1.001	0.997	1.005	0.734
PET-positive findings (yes vs. no)	1.225	0.769	1.952	0.394
Pelvic lymph nodes in PET (yes vs. no)	1.035	0.651	1.645	0.885
Distant metastases in PET (yes vs. no)	1.520	0.886	2.606	0.128

Multivariate analysis was also conducted with PSA before PET as a dichotomous variable (cut-point: 0.5 ng) achieving similar results (Cox ratio: 1.05, 95% CI: 1.02–1.08, *p* = 0.003). Identical results were also observed for Cox regression analyses using pre-sRT PSA values instead of pre-PET PSA values. Bold: statistically significant values. CI: confidence interval.

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
