# Peer review of "Outcome After 68Ga-PSMA-11 versus Choline PET-Based Salvage Radiotherapy in Patients with Biochemical Recurrence of Prostate Cancer: A Matched-Pair Analysis"

_cancers, 2020, doi:10.3390/cancers12113395_

Round 1
Reviewer 1 Report
The authors performed a large retrospective bi-center analysis of patients that received PSMA or Choline PET before salvage radiotherapy. The authors present first outcome data and comparison between both PET tracers. Overall, the paper deals with an interesting and important topic and the data is well presented. I have just two minor remarks regarding discussion of the results:
1)The higher local recurrence rate in the Choline PET group is a bit surprising given the lower sensitivity of this tracer. This might be a hint, that the high PET intensitiy of PSMA within the bladder might hamper correct detection of local recurrent diease. The authors should discuss this finding a bit more in detail.
2) The authors should also include other publications on outcome after PSMA based radiotherapy, although being sparse, there are some more publications e.g. PMID: 30488099 and 28830532
Author Response
The authors performed a large retrospective bi-center analysis of patients that received PSMA or Choline PET before salvage radiotherapy. The authors present first outcome data and comparison between both PET tracers. Overall, the paper deals with an interesting and important topic and the data is well presented. I have just two minor remarks regarding discussion of the results:
1)The higher local recurrence rate in the Choline PET group is a bit surprising given the lower sensitivity of this tracer. This might be a hint, that the high PET intensitiy of PSMA within the bladder might hamper correct detection of local recurrent diease. The authors should discuss this finding a bit more in detail.
Authors’ response: We would like to thank Reviewer 1 for this helpful comment. We mentioned this important issue in the discussion section.
Revision action, discussion (page 12): “A possible explanation for this observation could be the high bladder signal of the used 68Ga-HBED-CC PSMA tracer, which might hamper correct detection of local recurrent disease.”
2) The authors should also include other publications on outcome after PSMA based radiotherapy, although being sparse, there are some more publications e.g. PMID: 30488099 and 28830532
Authors’ response: We would like to thank Reviewer 1 for referring these two important works. We cited both in the discussion.
Reviewer 2 Report
Outcome after 68Ga-PSMA-11 versus choline 2 PET-based salvage radiotherapy in patients with 3 biochemical recurrence: a matched-pair analysis
By Nina-Sophie Schmidt-Hegemann, et al
In this paper, the biochemical-recurrence free survival (BRFS) after salvage radiotherapy (sRT) was analyzed using multivariate analysis, only PSA before PET imaging and PSA before sRT were significantly associated with BRFS. PSMA PET-based sRT had no superior BRFS rates when compared with choline PET-based sRT.
The paper is well written, and should be published.
Specific comments:
- Page 3, line 110, remove extra “,”
- The format of all reference should be checked and modified according to the requirements of the journal.
Author Response
Outcome after 68Ga-PSMA-11 versus choline 2 PET-based salvage radiotherapy in patients with 3 biochemical recurrence: a matched-pair analysis
By Nina-Sophie Schmidt-Hegemann, et al
In this paper, the biochemical-recurrence free survival (BRFS) after salvage radiotherapy (sRT) was analyzed using multivariate analysis, only PSA before PET imaging and PSA before sRT were significantly associated with BRFS. PSMA PET-based sRT had no superior BRFS rates when compared with choline PET-based sRT.
The paper is well written, and should be published.
Specific comments:
Page 3, line 110, remove extra “,”
Authors’ response: We would like to thank Reviewer 2 for this remark and we deleted the “,”
Reviewer 3 Report
Review cancers-980981: Outcome after 68Ga-PSMA-11 versus choline PET-based salvage radiotherapy in patients with biochemical recurrence: a matched-pair analysis
The authors took up a very interesting and hot topic of the comparison of the two imaging techniques and the assessment of their impact on biochemical progression after emergency radiotherapy in patients with prostate cancer. Importantly, there is a lack of data on this issue in the scientific literature.
General remarks are listed below:
Shouldn't the words "... of prostate cancer ..." appear in the title? - for intuitive topic identification
Unfortunately, the considerable heterogeneity of the analyzed cohort raises major concerns - You analyzed all patients who had sRT based on PET-CT imaging.
First of all, patients with persistent PSA levels after prostatectomy do not meet the criteria for biochemical recurrence - these are non-radical patients - so it is difficult to speak of a relapse here.
This contradicts the main assumption of the study and this group of patients should be excluded from this analysis, because it may have a significant impact on the presented results.
In the group with PSMA, patients with pN + constituted as much as 1/3 of the analyzed patients and in the group with choline only 12.7%. Patients with metastases before surgical treatment were excluded from the analysis, but what with pN+ patients ? Is cN+ - not a metastatic disease and it is known that the presence of this feature is significant on reccurence-free survival itself?
There are no differences in BRFS in the whole cohort analysis - why? Does this blur with the difference in severity to the disadvantage of patients with PSMA who have undergone extended nodal sRT therapy?
Thanks to above, do they have a non-inferior reccurence-free survival than patients in the choline group?
It seems that an ideal comparison would be patients who had a true biochemical recurrence after prostatectomy who had no nodal metastases. Naturally, such restrictions significantly reduce the study population, but homogeneity allows for meaningful conclusions to be drawn.
Matched-pair analysis is a valuable calculation, but once again the heterogeneity of the entire cohort creeps in, which, despite the lack of significant differences in group characteristics, probably overshadows the final conclusion.
Technical remarks:
- This journal don’t have strict formatting requirements, but the layout of the sections in this manuscript needs to be rebuilt: the reader should be familiar with the research materials and methodology before presenting the results and discussing it - which is, after all, a classic layout of a scientific article.
- Section 4.3. PET / CT Imaging should be presented before Treatment and follow-up according to time line of each section.
- References: Needs to be standardized. Items from the same journal are presented differently, i.e. ref 7 and 16
Author Response
The authors took up a very interesting and hot topic of the comparison of the two imaging techniques and the assessment of their impact on biochemical progression after emergency radiotherapy in patients with prostate cancer. Importantly, there is a lack of data on this issue in the scientific literature.
General remarks are listed below:
Shouldn't the words "... of prostate cancer ..." appear in the title? - for intuitive topic identification
Authors’ response: We would like to thank Reviewer 3 for this absolutely reasonable remark. We changed the title to “Outcome after 68Ga-PSMA-11 versus choline PET-based salvage radiotherapy in patients with biochemical recurrence of prostate cancer: a matched-pair analysis”
Unfortunately, the considerable heterogeneity of the analyzed cohort raises major concerns - You analyzed all patients who had sRT based on PET-CT imaging.
First of all, patients with persistent PSA levels after prostatectomy do not meet the criteria for biochemical recurrence - these are non-radical patients - so it is difficult to speak of a relapse here. This contradicts the main assumption of the study and this group of patients should be excluded from this analysis, because it may have a significant impact on the presented results.
Authors’ response: Reviewer 3 is right that we included two different groups of patients: with PSA persistence and with PSA recurrence. Indeed, several studies proposed that patients with PSA persistence have worse outcome. In line with the other studies, PSA persistence was also a significant factor for PSA relapse in our study (in analyses including all patients). This is why we have chosen PSA persistence as a parameter for propensity score matching of the cohorts. Nevertheless, we followed reviewer 3’s recommendations and we performed two additional analyses:
1 Cox-regression analysis after exclusion of patients with persistent PSA after surgery from the entire cohort: we observed comparable results to the entire cohort
2 Comparison of BRFS between PSMA-PET vs Cholin-PET patients for patients with PSA relapse and for patients with persistent PSA in the matched pairs cohorts: we observed comparable results to the entire cohort
Revision action 1, results (page 5): “Uni- and multivariate Cox-regression analyses were also performed by excluding all patients with PSA persistence after surgery. Thus, only patients with biochemical recurrent disease were considered (n=250) and comparable results were obtained (PSA values before PET imaging or sRT are the only significant factors in multivariate analyses).”
Revision action 2, results (page 6): “Furthermore, no statistically significant differences between patients with PSMA PET-based vs. choline PET-based sRT was observed by considering only patients with persistent PSA levels (p=0.343) or only patients with PSA relapse (p=0.345) after surgery, respectively.”
In the group with PSMA, patients with pN + constituted as much as 1/3 of the analyzed patients and in the group with choline only 12.7%. Patients with metastases before surgical treatment were excluded from the analysis, but what with pN+ patients ? Is cN+ - not a metastatic disease and it is known that the presence of this feature is significant on reccurence-free survival itself?
Authors’ response: Reviewer 3 is right that patients with pN+ have significant worse outcome compared to pN- patients. This was also reproducible in our analysis since pN+ was a significant factor in univariate Cox-regression analysis including the entire cohort. This is the reason why we considered pN status as a factor for propensity score matching. In the matched-pair analysis pN+ and pN- patients were equally distributed between the cholin and PSMA PET cohorts (please see supplementary table 1 and results 3.3). Thus, we believe that this factor is negligible due to the design of our study (pairwise comparison after propensity score matching). However, under consideration of the Reviewer’s suggestion we performed a sub-group analysis including only patients with biochemical recurrent disease and pN- status (please see the following).
We did not exclude patients with cN+ disease in PET in order to ensure a fair comparison between cholin PET and PSMA PET patients. Previous studies suggested that PSMA PET has a higher detection rate (also for lymph nodes) and this was also observed in our work. Based on the detection of positive lymph nodes a change in treatment strategy was performed: ADT, extended fields to the pelvic lymphatics and boost to the positive lymph nodes. An exclusion of the cN+ patients would have impaired the comparison between both tracers since a possible advantage of PSMA PET imaging (initiation of aggressive therapies in more patients due to detection of more positive lymph nodes) would have been diminished.
There are no differences in BRFS in the whole cohort analysis - why? Does this blur with the difference in severity to the disadvantage of patients with PSMA who have undergone extended nodal sRT therapy? Thanks to above, do they have a non-inferior reccurence-free survival than patients in the choline group?
Authors’ response: We compared BRFS in patients with cholin vs PSMA PET before sRT in two ways: before and after propensity score matching. Before propensity score matching cholin PET patients had significant better BRFS which is easily explainable by worse clinical parameters of the PSMA PET patients. To account for this, we also performed propensity score matching between the two PET tracer groups (PSMA vs. Choline) based on pre-PET PSA levels, pathological lymph node status, postoperative PSA persistence vs. PSA recurrence and NCCN risk groups and a matched patient cohort was created. Afterwards no significant difference in BRFS between PSMA PET and cholin PET patients was observed. There are several explanations for this phenomenon which we discussed in the current form of the manuscript: “A possible explanation for this finding is that the current intensification of treatment strategies based on PET-findings (e.g. administration of ADT or additional treatment of pelvic lymphatic pathways) are so far not sufficient enough. Additionally, the retrospective design of our study could have led to a heterogeneous distribution of prognostic relevant clinical parameters between both patient groups which may not have been accounted for by propensity score matching.
It seems that an ideal comparison would be patients who had a true biochemical recurrence after prostatectomy who had no nodal metastases. Naturally, such restrictions significantly reduce the study population, but homogeneity allows for meaningful conclusions to be drawn.
Authors’ response: Reviewer 3 is right. Homogeneity is important to draw meaningful conclusions. We also performed the analysis by excluding patients with pN+ and patients with PSA persistence to obtain only patients with true biochemical recurrence after prostatectomy.
Revision action, results (page 6): “Finally, only patients with biochemical recurrent disease and without lymph nodes in surgery (pN0 status) were analyzed (n=180) and again no statistically significant difference between patients with PSMA PET-based vs. choline PET-based sRT was observed (p=0.448)”
Revision action, discussion (page 12): “However, we also compared choline- versus PSMA-based sRT in patients with true biochemical recurrence after surgery (by excluding patients with PSA persistence and patients with pN+ status) and did not find any statistically significant differences.”
Matched-pair analysis is a valuable calculation, but once again the heterogeneity of the entire cohort creeps in, which, despite the lack of significant differences in group characteristics, probably overshadows the final conclusion.
Authors’ response: Reviewer 3 is right. We addressed this issue in the discussion.
Revision action, discussion (page 12): “Additionally, the retrospective design of our study could have led to a heterogeneous distribution of prognostic relevant clinical parameters between both patient groups which may not have been accounted for by propensity score matching.”
Technical remarks:
- This journal don’t have strict formatting requirements, but the layout of the sections in this manuscript needs to be rebuilt: the reader should be familiar with the research materials and methodology before presenting the results and discussing it - which is, after all, a classic layout of a scientific article.
Authors’ response: We re-arranged the manuscript as Reviewer 3 proposed.
- Section 4.3. PET / CT Imaging should be presented before Treatment and follow-up according to time line of each section.
Authors’ response: We re-arranged the manuscript as Reviewer 3 proposed.
- References: Needs to be standardized. Items from the same journal are presented differently, i.e. ref 7 and 16
Authors’ response: We standardized the references.
Reviewer 4 Report
Schmidt-Hegemann et al. present a bi-institutional retrospective study, whose aim was to analyze biochemical-recurrence free survival (BRFS) after PET-guided salvage radiotherapy (sRT) in a large cohort of 421 patients and at a median follow-up of 30 months. The main findings of the study are the following: 1. PSA levels prior sRT were confirmed to be a strong prognostic effect, 2. PET results were predictive of BRFS with overall lower BRFS in patients with PET-positive disease not confined to the prostatic fossa, and 3. the comparison between PSMA vs. choline PET/CT-based sRT showed that, despite the higher detection rate of PSMA-PET, BRFS rates were independent of the PET tracer applied.
This is an interesting study. The research question is well defined and clinically relevant. The conclusions are in concordance with the presented results. The literature cited is up-to-date and the article is well-written. On the other hand, a significant limitation of the study lies in the different treatment protocols applied in patients, which introduces a degree of bias in the statistical analysis.
The authors should address the following points:
Abstract
Adequate, no remarks.
Introduction
Adequate, no remarks.
Materials and methods:
- Were SUV calculations performed? It would be very interesting to investigate the potential impact of tracers’ uptake on BRFS.
Results:
- Could the authors provide some more detailed data regarding 18F- and 11C-choline PET/CT? Were any differences observed between the two tracers regarding PCa detection rate?
- Would the authors consider providing the respective Kaplan-Meier curves for PET-positive vs. PET-negative patients, in order to highlight the impact of PET imaging on survival?
- The authors mention that the two PET-based patient groups (PSMA vs. choline) were well balanced regarding pre-PET PSA levels, NCCN risk groups, pN stage and PSA persistence after surgery (ln 152-153) but not for pT stage, R status and, importantly, for the treatment protocols applied, which introduce a degree of bias. How did the authors attempt to tackle/reduce this effect?
Discussion:
- In line with results of previous studies in the field (Emmett 2019; Giovacchini 2019), the authors show that patients with PET-positive findings had significantly worse BRFS compared to patients without PET-positive findings. At the same time however, the 2- and 3-year BRFS are similar to several studies reporting on non-PET guided sRT, as nicely mentioned in the paper (ln 179-180; references 2, 25-29). Moreover, the impact of PET findings on survival was not confirmed in the multivariate analysis. Based on the previous, one could question the clinical benefit of introducing an expensive and carrying a radiation burden modality in the management of these patients. How would the authors address this argument?
Tables and Figures:
Table 3: Needs some modifications, especially the separation between uni- and multivariate Cox-regression analysis should be more marked.
Author Response
Schmidt-Hegemann et al. present a bi-institutional retrospective study, whose aim was to analyze biochemical-recurrence free survival (BRFS) after PET-guided salvage radiotherapy (sRT) in a large cohort of 421 patients and at a median follow-up of 30 months. The main findings of the study are the following: 1. PSA levels prior sRT were confirmed to be a strong prognostic effect, 2. PET results were predictive of BRFS with overall lower BRFS in patients with PET-positive disease not confined to the prostatic fossa, and 3. the comparison between PSMA vs. choline PET/CT-based sRT showed that, despite the higher detection rate of PSMA-PET, BRFS rates were independent of the PET tracer applied.
This is an interesting study. The research question is well defined and clinically relevant. The conclusions are in concordance with the presented results. The literature cited is up-to-date and the article is well-written. On the other hand, a significant limitation of the study lies in the different treatment protocols applied in patients, which introduces a degree of bias in the statistical analysis.
The authors should address the following points:
Abstract
Adequate, no remarks.
Introduction
Adequate, no remarks.
Materials and methods:
- Were SUV calculations performed? It would be very interesting to investigate the potential impact of tracers’ uptake on BRFS.
Authors’ response: We fully agree with Reviewer 4 that a correlation between PET-imaging features and BRFS would be very interesting. However, due to the heterogeneity of the used tracers and the used scanners in the current work we did not perform this meaningful correlation. However, in an ongoing project we are investigating the impact of PSMA PET radiomic features on BRFS by including also other patients from other centres to increase the statistical power of the analysis. We stated that this analysis should be done in the discussion of the manuscript.
Revision action, results (page 12): “In this context, future studies should also address whether radiomic features derived from PET images 45 may improve risk stratification models for patients undergoing PET-guided sRT.”
Results:
- Could the authors provide some more detailed data regarding 18F- and 11C-choline PET/CT? Were any differences observed between the two tracers regarding PCa detection rate?
Authors’ response: We thank Reviewer 4 for suggesting this interesting analysis. We compared the detection rates and we did not found any differences between the two tracers. However, we decided to not implement this additional information in the final version of the manuscript since the main question of this paper was to compare BRFS between PSMA PET and cholin PET. We believe that the usage of choline PET will decrease in the future and that this additional information would blur the focus of the readers.
- Would the authors consider providing the respective Kaplan-Meier curves for PET-positive vs. PET-negative patients, in order to highlight the impact of PET imaging on survival?
Authors’ response: To highlight the impact of PET imaging on BRFS we implemented the new Figure 2 in the manuscript
- The authors mention that the two PET-based patient groups (PSMA vs. choline) were well balanced regarding pre-PET PSA levels, NCCN risk groups, pN stage and PSA persistence after surgery (ln 152-153) but not for pT stage, R status and, importantly, for the treatment protocols applied, which introduce a degree of bias. How did the authors attempt to tackle/reduce this effect?
Authors’ response: We would like to thank Reviewer 4 for this important remark.
We did not account for the R status during the propensity score matching since it was not a significant variable in the univariate Cox regression. We also did not account for the pT stage (although it was significant in the univariate analysis) since 9.6% of the patients in the choline group had unknown pT status. However, the pT stage was well balanced between the PSMA and choline patients: exactly 27.9% had pT3a and 21.3% (choline) / 22.1 (PSMA) had pT3b. In the choline group more patients had pT4 (3.7% vs 1.5%).
The treatment protocols did not change over the time. Thus, the sRT protocols were identical between the choline PET and PSMA PET patients. We clarified this important issue in the material and methods section.
Revision action, materials and methods (page 4): “SRT concepts were identical between the patients which received choline and PSMA PET regarding RT technique, treatment fields, administration of ADT and overall doses and have been previously published 24.”
Discussion:
In line with results of previous studies in the field (Emmett 2019; Giovacchini 2019), the authors show that patients with PET-positive findings had significantly worse BRFS compared to patients without PET-positive findings. At the same time however, the 2- and 3-year BRFS are similar to several studies reporting on non-PET guided sRT, as nicely mentioned in the paper (ln 179-180; references 2, 25-29). Moreover, the impact of PET findings on survival was not confirmed in the multivariate analysis. Based on the previous, one could question the clinical benefit of introducing an expensive and carrying a radiation burden modality in the management of these patients. How would the authors address this argument?
Authors’ response: We would like to thank Reviewer 4 for opening this interesting discussion. Considering the heterogeneous study populations and treatment protocols the direct comparison between our results and the results of non-PET sRT studies is massively hampered. The only way to properly answer the question whether PET-guided sRT improves the oncologic outcome is the conduction of prospective trials which are currently undergoing. Consequently, we really don’t know whether the implamentation of PET in sRT planning is of oncological benefit, especially, under consideration of the costs and the radiation burden. We discussed this issue in the new version of the manuscript.
Revision action, discussion (page 11): “Trials explicitly investigating the oncologic benefit of e.g. PSMA PET-based sRT are currently underway but results are still pending (NCT03582774) 34. Until then it remains unclear whether implementing PET/CT as an expensive and radiation burden carrying modality improves the management of these patients.”
Tables and Figures:
Table 3: Needs some modifications, especially the separation between uni- and multivariate Cox-regression analysis should be more marked.
Authors’ response: We clearer separated both analyses in the new version of Table 3.

Round 2
Reviewer 3 Report
Thank you for the comprehensive answers and explanations.
The corrections made to the manuscript dispel scientific doubts.
Reviewer 4 Report
The authors have addressed all comments.